# Molecular Characterization of Dehydrin in Azraq Saltbush among Related *Atriplex* Species

**DOI:** 10.3390/biotech12020027

**Published:** 2023-04-07

**Authors:** Anas Musallam, Saeid Abu-Romman, Monther T. Sadder

**Affiliations:** 1Biotechnology Research Directorate, National Agricultural Research Center, Baq’a 19381, Jordan; anas_musllam@yahoo.com; 2Department of Biotechnology, Faculty of Agricultural Technology, Al-Balqa Applied University, Al-Salt 19117, Jordan; saeid.aburomman@bau.edu.jo; 3Plant Biotechnology Lab, Department of Horticulture and Crop Science, School of Agriculture, University of Jordan, Amman 11942, Jordan

**Keywords:** *Atriplex*, dehydrin, LEA II, K–segment, S–segment

## Abstract

*Atriplex* spp. (saltbush) is known to survive extremely harsh environmental stresses such as salinity and drought. It mitigates such conditions based on specialized physiological and biochemical characteristics. Dehydrin genes (*DHNs*) are considered major players in this adaptation. In this study, a novel *DHN* gene from Azrak (Jordan) saltbush was characterized along with other *Atriplex* species from diverse habitats. Intronless *DHN*-expressed sequence tags (495–761 bp) were successfully cloned and sequenced. Saltbush dehydrins contain one S-segment followed by three K-segments: an arrangement called SK3-type. Two substantial insertions were detected including three copies of the K2-segemnet in *A. canescens*. New motif variants other than the six-serine standard were evident in the S-segment. AhaDHN1 (*A. halimus*) has a cysteine residue (SSCSSS), while AgaDHN1 (*A. gardneri var. utahensis*) has an isoleucine residue (SISSSS). In contrast to the conserved K1-segment, both the K2- and K3-segment showed several substitutions, particularly in AnuDHN1 (*A. nummularia*). In addition, a parsimony phylogenetic tree based on homologs from related genera was constructed. The phylogenetic tree resolved DHNs for all of the investigated *Atriplex* species in a superclade with an 85% bootstrap value. Nonetheless, the DHN isolated from Azraq saltbush was uniquely subclustred with a related genera *Halimione portulacoides*. The characterized DHNs revealed tremendous diversification among the *Atriplex* species, which opens a new venue for their functional analysis.

## 1. Introduction

Salinity is a major global stress factor affecting modern agriculture, especially in arid and semi-arid regions [1,2,3]. Enormous agricultural losses are caused by salinity and are expected to increase in the coming years. Saltbush (*Atriplex* spp.) belongs to the Chenopodiaceae family, which also encompasses major crops, e.g., spinach, sugar beet, and quinoa. It has several ploidy levels, e.g., 2n = 18, 36, 54, 72, or 90 [4]. Different *Atriplex* spp. are native to different regions, e.g., Eurasian red orache (*A. prostrate*) [5], American Gardner’s saltbush (*A. gardneri*) [6], Australian oldman saltbush (*A. nummularia*) [7], and Mediterranean saltbush (*A. halimus*) [2]. The latter spreads in a major desert region in Jordan called Azraq, which has aridisols with prevailing dry hot summers and dry cold winters (Figure 1) (personal communications).

Saltbush is a halophyte which possesses unique salt tolerance mechanisms that guarantee normal growth and development, e.g., *A. halimus* can grow in extreme sodic soils (25–30 dS m^−1^) (Le Houérou 1992). Moreover, moderate salinity inhibitory to major crops [1,8,9,10] would stimulate growth in *A. nummularia* and *A. halimus* [2,11].

Many efforts have been directed toward the characterization of relevant salt-responsive genes underlying the unique physiology of halophyte plants [12,13]. In this regard, saltbush would sense salinity stress or even drought stress through a specialized protein, which was previously called early responsive to dehydration (ERD). The new name for ERD is the osmosensitive calcium-permeable cation channel (OSCA). The OSCA was first reported in Arabidopsis by two separate research groups [14,15]. In saltbush, *AhOSCA* expression was found to be upregulated ca. 6-folds in *A. halimus* under 150 mM NaCl level (ca. 13 dS m^−1^), which mimics the salinity levels in the Azraq region [2].

*Dehydrins* (*DHNs*) are widely spread across the plant kingdom and are considered crucial stress-responsive genes, e.g., *FcDHN* in common fig under salinity stress [9] and tomato TAS14 dehydrin under drought stress [16]. DHNs contain hydrophilic residues that have functional changes in response to solutes and dehydration [13,17]. DHNs are members of the Late Embryogenesis Abundant II (LEA II) protein family, which are considered stress-responsive factors and are predominant during seed maturation and dissection phases [17]; in addition, they are considered to be major salinity tolerance biomarkers [13]. Nonetheless, there are other important salinity biomarkers, and some overlap even with drought stress but others do not [16].

DHN has three conserved segments in its structure (Y, S, and K segments), where K segments can be available once, twice, or trice forming amphipathic α-helices, which aid in stabilizing cellular components and membranes [18]. Both α-amylase and lactate dehydrogenase are examples of cold-sensitive enzymes protected by DHNs [19]. Additionally, DHNs can protect against oxidative stress [20]. Different *Atriplex* species are widely distributed around the world with diverse prevailing environmental conditions. Therefore, a colinearity is expected between the protein structure and function for each species, which would be vital for adaptation. Therefore, this study aimed to characterize DHN from Azrak (Jordan) saltbush among a group of diverse *Atriplex* species at the molecular level.

## 2. Materials and Methods

### 2.1. Plant Materials

The saltbush seeds were collected from Azrak (Jordan). In addition, seeds from different *Atriplex* spp. were kindly provided from two seed banks: the National Agricultural Research Center (NARC, Jordan) and the National Arid Land Plant Genetic Resource Unit (USDA, USA). The seeds were surface-sterilized and germinated in vitro over MS medium. The plantlets were subcultured using nodal cutting [13]. A total of fourteen accessions covering nine different species were included in the study (Table 1): thirteen *Atriplex* spp. accessions and *Halimione portulacoides* (sea purslane), a sister species that was formerly classified as *A. portulacoides*.

### 2.2. Cloning DHNs

Total RNA was extracted from fresh leaves based on the guanidinium thiocyanate phenol—chloroform method [20] using TRIZOL (Invitrogen Inc, USA). All used plastic-wares were RNase free, and RNA was resuspended in DEPC-treated water (Qiagen, Germany) supplemented with an RNase inhibitor (Qiagen, Germany). Reverse transcription reactions were performed for isolated RNA templates following the recommended procedure using a GoScript™ Reverse Transcriptase kit (Promega, USA). Several primers were designed to amplify ESTs covering the majority of DHN genes using the polymerase chain reaction (online Appendix A). The fragments were cloned as described earlier [13] and sequenced with the Sanger method using an ABI 3730 sequencer (ABI, USA). The sequences were deposited in GenBank [21] with nucleotide accession numbers (MH591427-MH591440) and corresponding protein accession numbers (AYH52682-AYH52695) (Table 1).

### 2.3. Analysis of Atriplex DHNs

Protein secondary structure was predicted using Phyre2 software [22]. Protein features and predicted 3-dimensional structure were analyzed through the PSIPRED server hosted by University College London [23]. The three-dimensional structure with ligand prediction was predicted using I-TASSER software [24]. DHN proteins from related Chenopodiaceae were retrieved from Genbank [21]. They included proteins from two *Atriplex* species, *Chenopodium quinoa*, *Beta vulgaris*, *Spinacea oleracea,* and two *Suaeda* species (*salsa* and *glauca*) in addition to *Tamarix hispidata* as an outgroup. Protein sequences were subjected to multiple sequence alignment, along with saltbush dehydrin protein sequences generated in this study using BioEdit [25]. Aligned sequences were bootstrapped 1000 times by using the SEQBOOT function available in PHYLIP software [26] followed by the construction of a phylogenetic tree based on the parsimony method. A consensus tree was illustrated using TreeView [27].

## 3. Results

Seedlings were successfully grown for all *Atriplex* species. Thereafter, nodal cuttings were used to multiply and maintain the cultures in 250 glass bottles to obtain enough tissue materials for RNA isolation. The sequenced *DHN* ESTs were found to be in the range of 495-761 bp. Nonetheless, they cover all important segments (S and K).

When the DHN amino acid sequences were multiple-aligned, two major insertions were detected. The first insertion was short (KHETLGQ) and was detected in DHN protein from *A. canescens* (AFC98463), which was located 14 amino acids upstream of the S-segment. The second insertion detected 9 amino acids upstream of the K2-segment. Two different segments in the second insertion were evident; the first segment (Fifty amino acids long) has an additional K2-segment and was detected in *A. halimus* (AGZ86543), *A. dimorphostegia* (AYH52684), *A. leucoclada* (AYH52687), *A. hortensisi* (AYH52688), and *A. gardneri* var. *utahensis* (AYH52691). Interestingly, *A. numimularia* (AYH52689) showed a similar fifty amino acid insertion but without an additional K2-segment as above. The second segment (one hundred amino acids long) has two extra K2-segments, and it was detected only in *A. canescens* (AFC98463).

A membrane pore-lining stretch (14-6 amino acids) was predicted for all of the investigated DHNs (online Appendix A), where the N- and C-termini were cytosolic and extracellular, respectively, except for *A. nummularia*, which had two DHNs; AYH52689 had an extracellular N-terminus and a pore-lining stretch of EDAVISGVEKAHVFS (Figure 2A), while AYH52690 had a cytosolic N-terminus and a pore-lining stretch of HEAVT HVATAEPSVEG (Figure 2B). The secondary structure prediction for the investigated *Atriplex* DHNs was carried out for polypeptides spanning [N-terminus—S-segment—K1-segment] (online Appendix A). Accordingly, *Atriplex* DHNs showed unique helices with diverse numbers and lengths.

Concerning the conserved segments in DHN proteins, most species showed six residues of the conserved S-segment (SSSSSS) (Figure 3). However, *A. gardneri* var. utahensis (AYH52691) has the isoleucine (I) residue at the second position, and *A. nummularia* (AYH52690) has the threonine (T) residue at the second position. Moreover, *A. nummularia* (AYH52689) showed three cysteine (CCC) residues, and *A. halimus* (AYH52686) showed one cysteine residue (C) at the third position (Figure 3). Analysis of aligned DNA sequences showed several SNPs in the S-segment (S1S2S3S4S5S6) for the investigated Atriplex species. In S1, a silent mutation (TCC to TCT) was found in three *A. halimus* accessions (KF578414, MH591427, and MH591431). A S2I missense mutation was found in *A. gardneri* var. utahensis (AYH52691) as a result of a codon change from the consensus AGC to ATC. In addition, the *A. nummularia* (AYH52690) showed a S2T variant due to a codon change from the consensus AGC to ACC. Moreover, sequential serine to cysteine substitutions (S2C, S3C, and S4C) were detected in *A. nummularia* (AYH52689) due to three SNPs in the consensus (AGC TCT AGC to TGC TGT TGT). In addition, the *A. halimus* (AYH52686) DHN showed a S3C variant due to a codon change from the consensus TCT to TGT.

The highly conserved K1-segment (KKKRKKEKKEKK) was evident for different Atriplex species and accessions (Figure 3). Three amino acid substitutions were found in *A. canescens* (AFC98463); K3R was due to a codon change from AAG > AGG; K5R and K8R, the last two substitutions, resulted from a codon change from AAG to AGG.

The DHNs for both *A. nummularia* (MH591434) and *A. dimorphostegia* (MH591429) showed silent SNPs in R4 in the K1-segment (AGA to AGG). Moreover, a silent mutation in E7 (GAA>GAG) was evident in the DHNs of three *A. halimus* accessions (KF578414, MH591427, and MH591431) and *A. nummularia* (MH591434). In addition, *A. nummularia* (MH591434) has a silent mutation in K9 resulting from AAA>AAG.

The K2-segment is the third major motif present in DHN proteins with a consensus of KKGGFLDK(V/I)KDK (Figure 3). However, it was KKGGFVEKIKDK in *A. canscence* (AFC98463), while it was KKVGFLDKVKDK in several species; *A. halimus* (AYH52685), *A. dimorphostegia* (AYH52684), *A. leucoclada* (AYH52687), *A. hortensis* (AYH52688), *A. gardneri* var. utahensis (AYH52691), and *Atriplex lindleyi subsp. conduplicata* (AYH52692). The KKGGFLDKIKDK variant was present in *A. halimus* (AYH52682) and *A. halimus* (AYH52686), while it was KKGGFVDKIKDK in the *A. halimus* (AYH52683) accession and KKGGFLGKIKDK in the previously published *A. halimus* (AGZ86543)as KKGGFVDKIKDK in the *A. halimus* (AYH52683) accession and KKGGFLGKIKDK in the previously published *A. halimus* (AGZ86543) accession. On the other hand, the DHN K2-segments of *A. nummularia* (AYH52689) and (AYH52690) were KVGGLDDVVEDL and KKGGFLDKVKDK, respectively.

When investigating the colinear nucleotide sequences covering the K2-segment, fourteen SNPs were detected in different positions. *A. nummularia* (AYH52689) had a K2V resulting from AAG > GTT and a silent SNP in G3 with GGT > GGA as compared with *A. canscence* (AFC98463), *A. halimus* (AGZ86543, AYH52682, AYH52686, and AYH52683), and *A. nummularia* (AYH52690). In addition, it had an F5L substitution with TTC > CTC and another one with L6D resulting from CTT > GAT as compared to all other species and accessions except for *A. canscence* (AFC98463) and *A. halimus* (AYH52683). On the contrary, an L6V (CTT > GTT) substitution was evident in *A. canscence* (JN974246) and *A. halimus* (AYH52683). Additional K2-segment substitutions in *A. nummularia* (AYH52689) compared to the consensus include: K8V (AAG > GTC), K10E (AAG > GAG), and K12L (AAA > CTC). Furthermore, the *DHNs* of *A. halimus* (AYH52685); *A. dimorphosegia* (AYH52684); *A. leucoclada* (AYH52687); *A. hortensis* (AYH52688); and *A. gardneri* var. *utahensis* (AYH52691) had G3V substitutions resulting from GGT > GTT, while V9I (GTC > ATC) was evident in *A. canscence* (AFC98463) and different *A. halimus* accessions (AGZ86543, AYH52682, AYH52686, and AYH52683).

The third K-rich segment in DHN Atriplex species is K3-segment. This segment is highly conserved and has consensus (KDKKGFLDKIKDKIPG) in all of the investigated species except *A. nummularia* (AYH52689) which has a special K3-segment (EEKEVFVEMIEDFSRQ) resulting from several SNPs; K1E (AAG > GAG), D2E (GAT > GAG), K4E (AAG > GAG), G5V (GGG > GTC), L7V (TTG > GTG); D8E (GAC > GAG), K9M (AAG > ATG), K11E (AAG > GAG), K13F (AAA > TTC), I14S (ATC > TCC), P15R (CCT > CGG), and G16Q (GGC > CAG). On the other hand, *A. halimus* (AYH52682) has two silent mutations in the K3-segment sequence at I14 and (ATC > ATT) P15 (CCT > CCC).

The three-dimensional structure was predicated for the DHN protein from *A. halimus* (AYH52683) grown in the Azraq region (Jordan). The structure contains eight long and three short helices (Figure 4). Three ligands were predicted to bind the protein, namely, Mg, Ca, and Fe.

A parsimonious phylogenetic tree was constructed for all available DHN proteins from the Atriplex species and the homologs from other species in the Chenopodiaceae (Figure 5). The Tamarix hispidata (out-group) DHN was nicely separated from all of the other DHNs. The tree showed two major clusters. The first one was resolved with a 98% bootstrap value. It comprises two DHNs from Chenopodium quinoa.

The second cluster was subdivided into seven major clades. Clade (I) has one, two, and three DHNs from Spinacia oleracea, C. quinoa, and Beta vulgaris, respectively while clade (II) has two DHNs each from S. oleracea (XP_021848510 and KNA16389), C. quinoa (XP_021774793 and XP_021752990), and Beta vulgaris (KMT01303 and XP_010690283). Clade (III) was resolved with a 93% bootstrap value and has three DHNs from C. quinoua: ERD 14-like (XP_021732246), DHN1 (AGM15308), and ERD 14-like (XP_021756500). The biggest clade (IV) was resolved with a 85% bootstrap value and has 15 Atriplex species and accession DHNs: *A. halimus* (AGZ86543), *A. halimus* (AYH52686), *A. halimus* (AYH52682), *A. gardneri* (AYH52694), *A. gardneri* (AYH52695), *A. nummularia* L (AYH52690), *A. nummularia* U (AYH52689), *A. hortensis* (AYH52688), *A. canescens* (AFC98463), *A. leucoclada* (AYH52687), *A. dimorphostegia* (AYH52684), *A. lindleyi* (AYH52692), *A. halimus* (AYH52685), *A. gardneri* var. *utahensis* (AYH52691), and *A. halimus* (AYH52683), in addition to one DHN from the related species *Halimione portulacoides* (AYH52693). On the other hand, the smallest clade (V) was resolved with a 98% bootstrap value and has two DHNs from S. oleracea (KMS95553 and XP_021846321), while clade (VI) was resolved with a 64% bootstrap value and has two DHNs for each of the S. oleracea and Suaeda spp. Finally, clade (VII) has two proteins from Beta vulgaris.

## 4. Discussion

We were able to amplify *DHN* genes from several *Atriplex* species both indirectly using cDNA and directly using genomic DNA as they were found to be intronless. Likewise, several *DHNs* from different plant species were found to be intronless, e.g., *Eucalyptus globulus* [28] and *Vigna radiate* [29]. Many plants’ stress-responsive genes are void of introns or interspaced with just a few ones, e.g., MYB transcription factor [30] and salinity-responsive genes [31]. Furthermore, stress-responsive genes with rapid change in expression levels were also found to be interspaced with limited introns, e.g., *Arabidopsis thaliana* [32]. Moreover, recent studies have shown that such intronless and limited intron-interrupted genes were putatively acquired by horizontal gene transfer from prokaryotes to Phragmoplastophyta long before the appearance of terrestrial plants [33,34].

We were interested in investigating *DHN* genes from saltbush from Azraq (Jordan) and different *Atriplex* species as they were directly involved in salinity tolerance [16,35]. This was achieved by utilizing available sequences for *DHN* genes: *A. halimus* and A. *canscence* both with complete ORF. A recombinant yeast-expressing *AcDHN* gene was found to tolerate salinity stress [35]. Likewise, the expression of the *AhDHN* gene was found to be upregulated by more than seven folds in *A. halimus* roots under salinity stress [13]. However, they code for proteins that vary in structure and size. While the *AhDHN* gene (KF578414) from Saudi Arabia isolate encodes a 26.8 kDa protein [13], the *AcDHN* gene (JN974246) encodes a 38.3 kDa protein [35]. When compared with orthologs from the related genus *Chenopodium quinoa*, a wider range in protein sizes is evident, i.e., 30, 34, 50, and 55 kDa [36].

DHNs vary in their cellular location (membrane or cytoplasmic). They can have multiple locations in the cell, e.g., wheat DHNs [37]. Nonetheless, all of the investigated *Atriplex* DHNs in this study showed a single predicted pore-lining (online Appendix A) in comparison to comparable stress-responsive proteins with two pore-linings, e.g., OePMP3 [10]. Most *Atriplex* DHNs showed an extracellular C-terminus and a cytoplasmic N-terminus, similar to the previously published AhDHN [16]. However, *Atriplex nummularia* has two DHNs; one has an extracellular C-terminus and a cytoplasmic N-terminus similar to the other investigated DHNs, while the other has an extracellular N-terminus and a cytoplasmic C-terminus. The putative pore-lining of *Atriplex* DHNs indicates that they have major hydrophobic domains and consequently facilitate the folding of this integral protein along the phospholipid bilayer membrane. Likewise, plant DHN homologs were found to be associated with cell membranes. Arabidopsis DHNs were found to bind aquaporin (AtPIP2B) indicating a potential role in maintaining the lipid association of the aquaporin hydrophobic transmembrane portion [38]. Moreover, some plant DHNs were found to bind abiotic stress-responsive proteins for protection and to enhance their activity, e.g., ERD14 which can bind to Phi9 GLUTATHIONE-S-TRANSFERASE9 and CATALASE under oxidative stress in Arabidopsis [39], GsPM30 interacting with receptors such as cytoplasmic kinase GsCBRLK under salinity in wild soybean [40], and MtCAS31 protecting leghemoglobin under drought stress in barrel medic [41].

The detected amino acid substitutions in the major segments in the *A. halimus* DHN (Azraq region, Jordan) would interfere with its interaction with other macromolecules. For example, it was found that the S-segment is a major phosphorylation domain in Arabidopsis DHN [42]. Moreover, K-segments are positively charged as they are rich in K residues; therefore, they can bind DNA as the phosphate backbone is negatively charged [43]. Moreover, the expression of saltbush *DHN* (Azraq region, Jordan) was assessed using qRT-PCR (data not shown). The results were almost comparable with earlier findings for *DHN* expression under salinity stress (Al-Jouf region, Saudi Arabia) [13]. The saltbush *DHN* (Azraq region, Jordan) showed around ten-fold upregulation in the root tissues rather than the shoots compared with the seven-fold upregulation in *DHN* (Al-Jouf region, Saudi Arabia) [13].

Two GO terms related to molecular function were revealed for *A. halimus* DHN (Azraq region, Jordan), namely “substrate-specific transporter activity” and “protein binding”, while it showed four GO terms related to biological processes, “protein localization to nucleus”, “nuclear import”, protein import”, and “protein targeting”. Likewise, the maize DHN was found localized to the nucleus [44]. In addition, “nuclear envelope” and “pore complex” were predicted as GO terms related to cellular localization. In fact, both H residues and K-segments are indispensable for binding phospholipids, a major component of the cellular membrane [45,46].

On the other hand, three metal–ligand binding sites were predicted in the DHN from *A. halimus* (Azraq region, Jordan) (Figure 4). The first was a Mg ligand binding site with residues D148 and D152. The second was a Ca ligand binding site with residues D148 and D152. Similarly, the Arabidopsis DHN was found to bind calcium, which was found to be directly proportional to phosphorylation [42]. Finally, there was a Fe ligand binding site, which was associated with residues D162 and K191. A similar affinity to Fe was recorded in *Vitis riparia* DHN1 [47]. Furthermore, DHNs binding metal ligands can act as reactive oxygen species scavengers, which would aid in plant mitigation under various stresses [43,45,46].

The investigated species have a common genus (*Atriplex*), which belongs to the Chenapodaiceae family and subfamily of Chenapodioideae, which contains both *Atriplex* and *Chenapodium* genera [21]. After constructing the phylogenetic tree, all of the investigated *Atriplex* species in this study: (*A. halimus* (AYH52683), A. gardneri var. utahensis (AYH52691), *A. halimus* (AYH52685), *A. leucoclada* (AYH52687), *A. dimorphostegia* (AYH52684), *A. hortensis* (AYH52688) *A. nummularia* U (AYH52689), *A. halimus* (AYH52682), *A. nummularia* L (AYH52690), and *A. halimus* (AYH52686)) and the previously published data (*A. canescen* (AFC98463) and (NCBI 2022) and *A. halimus* (AGZ86543) [16]), were resolved in two subclades consolidated into one major clade (IV) with an 85% bootstrap value. They were separated from all other homologs from related genera, e.g., Chenopodium, Spinacia, Beta, and Suaeda. However, *Halimione portulacoides* also appeared in *Atriplex* clade IV. *H. portulacoides* (previously classified as *A. portulacoides*) was recently recognized as a distinct genus based on extensive phylogenetic analysis [48].

On the contrary, in a previous study, *A. halimus* was clustered away from *A. dimorphostegia* based on the *atpB-rbcL* spacer sequence, while *A. nummularia* and *A. hortensis* were clustered together and away from *A. halimus* and *A. leucoclada* based on ITS sequences [48]. This could mean that the DHN protein sequence is more conserved among *Atriplex* spp than DNA-based sequences (e.g., the *atpB-rbcL* spacer and ITS sequences). *Atriplex* and *Chenapodium* were clustered together but away from *Beta vulgaris* (Subfamily Betoidieae) based on *trnL-F* and *rpl16* sequences [49], the ITS sequence [50], and the matK/trnK sequence [51]. Likewise, our data showed two groups of *Chenapodium* DHNs, one that was clustered with *Atriplex* in a superclade resolved with an 80% bootstrap value and another group clustered into two major clades with *Beta* and *Spinacia*.

Based on an extensive phylogenetic analysis using maximum likelihood and utilizing 59 protein-coding genes [52], a tight clustering was reported for both *Spinacia* and *Chenapodium* followed by *Beta* among 11 genera from the Chenopodiaceae family. Likewise, *Spinacia* and *Chenapodium* were found to cluster together based on the *atpB-rbcL* spacer sequence [48]. In addition, and based on the flowering locus (*FT*), orthologs were clustered together between *Beta* and *Chenopodium* [53]. These clusters agree with our phylogenetic analysis of DHNs.

The cloned *DHN* from *A. halimus* (MH591431) showed several SNPs compared with the published *DHN* genes from *A. halimus* (KF578414) and *A. canescens* (JN974246). In addition, the highly conserved S-segment available in *DHN* genes from *A. halimus* and *A. canescens* consists of six residues of the amino acid serine (SSSSSS) [16,17], while the S-segment in the *DHN* gene from *A. halimus* (AYH52686) was found to have C instead of S at the third position (SSCSSS). Moreover, the PCR amplified *DHN* gene from *A. nummularia* showed two bands using gel electrophoresis, which could be an indication of the presence of two paralogous dehydrins in this species. Multiple dehydrins have already been recorded for several plant species. This can start from two copies as in *Vitis vinifera* and can go up to fifteen copies as in *Malus domestica* [54].

The sequencing of both fragments from *A. nummularia* revealed novel forms in the S-segment. The first upper band in *A. nummularia* (AYH52689) with a size of 761 bp gave an S-segment with three cysteines and three serines (SCCCSS), while the second lower band in the same species *A. nummularia* (AYH52690) with a size of 495 bp gave an S-segment rich in serein residues with threonine residues at the second position (STSSSS). On the contrary, *A. gardneri* var. *utahensis* (AYH52691) showed another variant in the S-segment with isoleucine replacing a serine residue at the second position (SISSSS). Although the S-segment in the DHN is highly conserved, however our data revealed novel variants reported for the first time, which is consistent with earlier works showing other variants such as the glycine-containing S-segment (S2G) in EjDHN2 from *Eriobotrya japonica* [55], the S2N available in *A. thaliana* (CAA62449) or the S2G form present in DHN3 from *Coffea canephora* [56].

The SK-segment showed more variability between species because it extends for a longer stretch (40–42 residues) and due to the presence of a gap in the gene structure [48]. This can lead to changes in the position of the S-segment between 1-18, while the K-segment runs between 27–42 positions. The most frequent SKn-segment present in the available DHNs is the YnSKn form (85%), while the SKn form is less frequent. Nonetheless, additional very rare forms were also recorded for plant dehydrins, e.g., the SKKS form present in *Stellaria longipes* [57] and the SKKYKY form present in *Cerastium arcticum* [58]. It is worth mentioning that the K-segment was found to have a protective role against both biotic and abiotic stresses, e.g., in grapes [59].

## 5. Conclusions

In this study, we found that most DHNs from the Azraq desert have a unique protein structure and presumably function. This could enable saltbush plants to survive the prevailing harsh conditions in the desert. On the other hand, the related DHNs from other species also showed unique and novel motifs (e.g., the S-segment) that would make the original plant be adapted to the specific conditions they live in. Therefore, the obtained data could guide future work to resolve holistic DHN interactomes with membranes, DNA, and other proteins to identify the uniqueness of each protein which would aid in mitigating different *Atriplex* spp. to different environmental stresses.

## Figures and Tables

**Figure 1 biotech-12-00027-f001:**
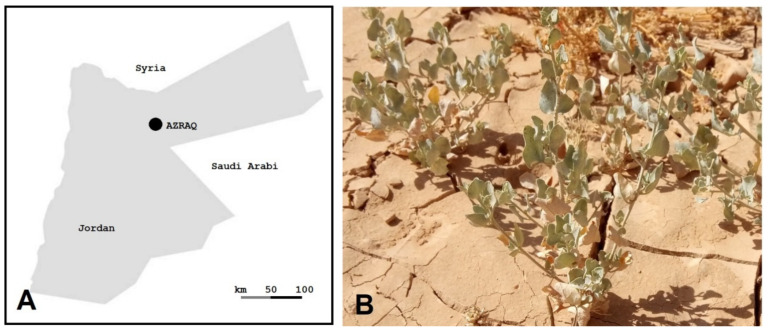
Saltbush native to Azraq. (**A**) A map of Jordan showing the Azraq region. (**B**) Saltbush growing in nature (Azraq) under dry and saline conditions; picture taken in mid-June 2022.

**Figure 2 biotech-12-00027-f002:**
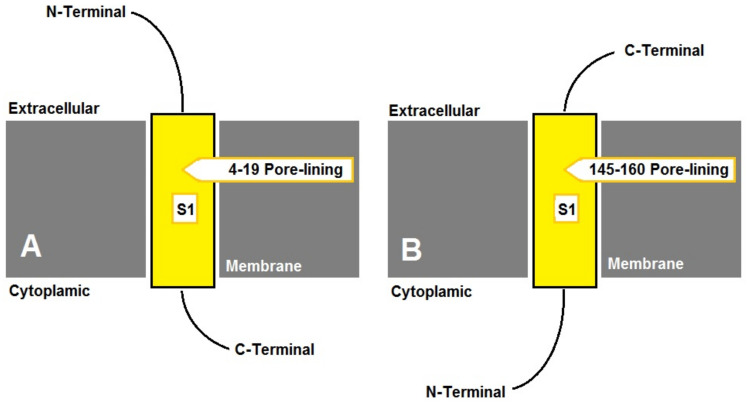
Predicted transmembrane topology for AnuDHN showing pore-lining amino acids: (**A**) for *Atriplex nummularia* (AYH52689); (**B**) for *Atriplex nummularia* (AYH52690).

**Figure 3 biotech-12-00027-f003:**
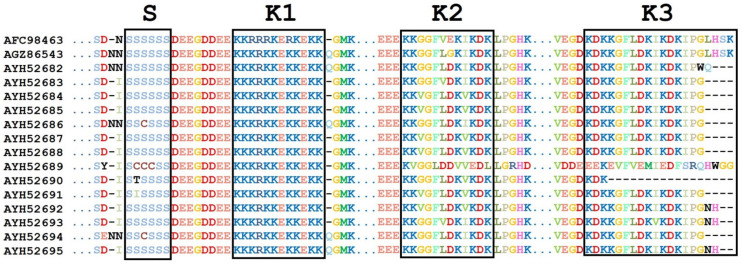
Multiple sequence alignment for DHN protein segments. AFC98463. (*A. canescens*), AGZ86543 (*A. halimus*), AYH52682 (*A. halimus*), AYH52683 (*A. halimus*), AYH52684 (*A. dimorphostegia*), AYH52685 (*A. halimus*), AYH52686 (*A. halimus*), AYH52687 (*A. leucoclada*), AYH52688 (*A. hortensis*), AYH52689 (*A. nummularia* U), AYH52690 (A. *nummularia* L), AYH52691 (A. *gardenri* var. utahensis), AYH52692 (A. *lindleyi* subsp. conduplicata), AYH52693 (*Halimione portulacoides*), AYH52694 (Atriplex *gardneri* U), and AYH52695 (*Atriplex gardneri* L).

**Figure 4 biotech-12-00027-f004:**
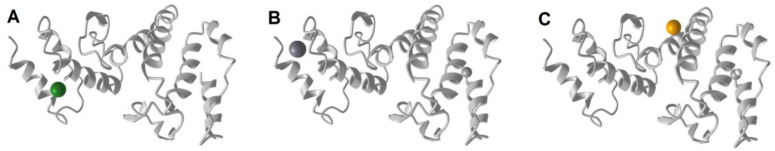
Predicted three-dimensional structure of the DHN protein (AYH52683) from *A. halimus*. along with the predicted ligand biding sites: (**A**) Mg, (**B**) Ca, and (**C**) Fe.

**Figure 5 biotech-12-00027-f005:**
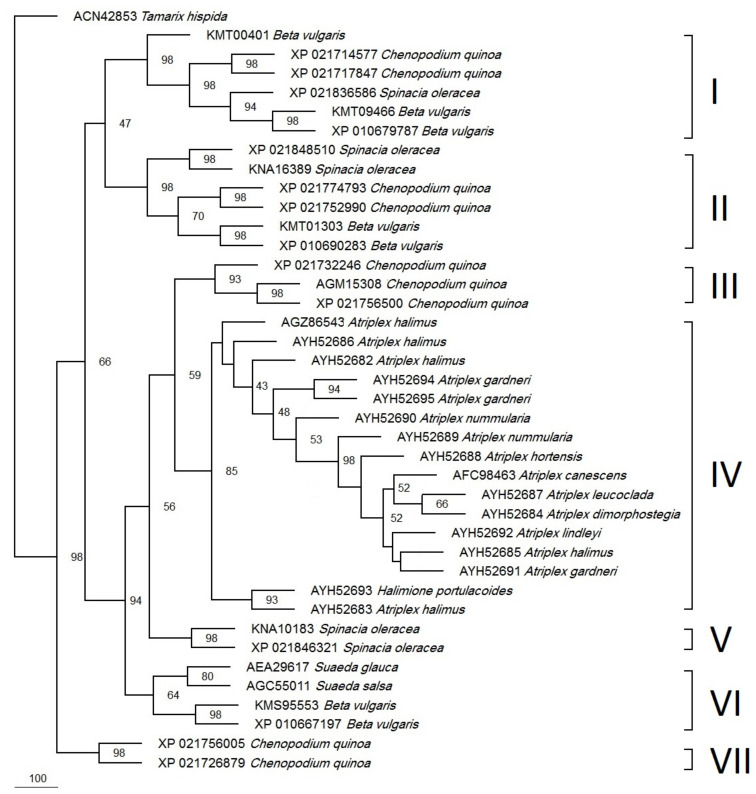
Phylogeny of the *Atriplex* DHN and homologs from related plant species based on the parsimony method. The accession numbers are indicated to the left of the scientific names. The branching points show the bootstrap values (percentage of 1000 runs). Clade I contains DHNs with SK2-type while the remaining clades contain DHNs with SK3-type.

**Table 1 biotech-12-00027-t001:** *Atriplex* species used in this study along with their source, lab code, DHN gene, and accession numbers.

#	Species	Source	Lab Code	Gene	EST Accession	Protein Accession
1	*Atriplex canescens*	NCBI	-	*AcaDHN1*	JN974246	AFC98463
2	*Atriplex halimus*	NCBI	-	*AhaDHN1*	KF578414	AGZ86543
3	*Atriplex halimus*	Amman, Jordan	JO372	*AhaDHN1*	MH591427	AYH52682
4	*Atriplex halimus*	Azraq, Jordan	JO2991	*AhaDHN1*	MH591428	AYH52683
5	*Atriplex dimorphostegia*	Aqaba, Jordan	JO3111	*AdiDHN1*	MH591429	AYH52684
6	*Atriplex halimus*	Al Jouf, Saudi Arabia	KSA	*AhaDHN2*	MH591430	AYH52685
7	*Atriplex halimus*	Bir Seb’a, Palestine	I4	*AhaDHN1*	MH591431	AYH52686
8	*Atriplex leucoclada*	Al-Naqab, Palestine	I5	*AleDHN1*	MH591432	AYH52687
9	*Atriplex hortensis*	Former Serbia and Montenegro	I15	*AhoDHN1*	MH591433	AYH52688
10	*Atriplex numimularia*	South Africa	I18U	*AnuDHN1*	MH591434	AYH52689
			I18L	*AnuDHN2*	MH591435	AYH52690
11	*Atriplex gardneri* var. *utahensis*	USA	I19	*AgaDHN1*	MH591436	AYH52691
12	*Atriplex lindleyi subsp. conduplicata*	South Africa	I6	*AliDHN1*	MH591437	AYH52692
13	*Halimione portulacoides*	Egypt	E2	*HpoDHN1*	MH591438	AYH52693
14	*Atriplex gardneri*	USA	I11U	*AgaDHN1*	MH591439	AYH52694
			I11L	*AgaDHN2*	MH591440	AYH52695

## Data Availability

Not applicable.

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
