# Peer review of "Molecular Characterization of Dehydrin in Azraq Saltbush among Related Atriplex Species"

_biotech, 2023, doi:10.3390/biotech12020027_

Round 1

Reviewer 1 Report

In this manuscript (biotech-2324023) entitled "Molecular characterization of dehydrin in Azraq saltbush among related Atriplex species" submitted to BioTech, authors characterized PqMYBF1, a novel DEHYDRIN (DHN) gene from Azrak (Jordan) saltbush along with other Atriplex species from diverse habitats. The data are convincing and the writing is clear and straightforward. However, some issues need to be addressed for improving the quality of this manuscript.

1, The growth and habitat of Azrak (Jordan) saltbush studied in this manuscript should be displayed as revised Figure 1. This new figure 1 is essential for author to understand the importance and novelty of this study, please provide in the revision

2, The expression pattern of DHN gene should be characterized in Azrak (Jordan) saltbush, especially it response to salinity stress. Please provide in the revision

3, Authors introduced the amino acid differences among DHN proteins from Atriplex species (e.g. page 5), but their biological significance should be discussed in the revised manuscript.

4, For Figure 4, difference in the binding sites for three ligands Mg, Ca and Fe should be discussed in the revision.

5, For Figure 5, bootstrap numbers should be provided in the revised legend. In addition, please show the domain arrangement for each proteins in the revised figure 5.

Author Response

-- We would like to thank the reviewer for the tremendous efforts and important comments and suggestions to improve the manuscript.

In this manuscript (biotech-2324023) entitled "Molecular characterization of dehydrin in Azraq saltbush among related Atriplex species" submitted to BioTech, authors characterized PqMYBF1, a novel DEHYDRIN (DHN) gene from Azrak (Jordan) saltbush along with other Atriplex species from diverse habitats.

The data are convincing and the writing is clear and straightforward.

-- Thank you for your positive comments.

However, some issues need to be addressed for improving the quality of this manuscript.

1, The growth and habitat of Azrak (Jordan) saltbush studied in this manuscript should be displayed as revised Figure 1. This new figure 1 is essential for author to understand the importance and novelty of this study, please provide in the revision

-- Thank you for your invaluable suggestion. The desert habitat with dry and saline soil was illustrated in new figure I showing saltbush in Azrak (Jordan)

2, The expression pattern of DHN gene should be characterized in Azrak (Jordan) saltbush, especially it response to salinity stress. Please provide in the revision

-- We already investigated the expression of DHN and other genes in Azrak (Jordan) saltbush under salinity in both roots and shoots. The results agrees with earlier findings, therefore, we did not include the qPCR. However, it was included in the revised manuscript as requested.

3, Authors introduced the amino acid differences among DHN proteins from Atriplex species (e.g. page 5), but their biological significance should be discussed in the revised manuscript.

-- A new paragraph was introduced to the discussion section. The importance of DHN sequence as related to phosphorylation and DNA binding were discussed and related literature were cited.

4, For Figure 4, difference in the binding sites for three ligands Mg, Ca and Fe should be discussed in the revision.

-- A new paragraph was introduced to the discussion section. The importance of DHN binding to cation ligands and the related GO terms were discussed and related literature were cited.

5, For Figure 5, bootstrap numbers should be provided in the revised legend. In addition, please show the domain arrangement for each proteins in the revised figure 5.

-- The bootstrap issue was clarified in legend, Clade I SK2  and all other clades SK3

Reviewer 2 Report

Comments for Authors

Abstract:

1.       There are lot of spelling mistakes typos. Please proof-read.

2.       SK3-type what? It needs to be specified.

3.       The examples of S-segment variations are not needed in abstract in my opinion.

Introduction:

1.       Make sure all scientific names are italicized.

2.       Some sentence structures are awkward and not clear due to the use of incorrect conjunctions. Please fix these.

3.       What happens in ERD? Is it a phase or response where genes are activated due to stress. Need to clarify this and tie it to the OSCA gene. Is OSCA the only gene that is affected by salinity? Or is it drought? or both?

4.       Is 150 mM NaCl equivalent to salinity in the environment?

5.       Are OSCA a type of dehydrin? There is no connection built in jumping from OSCA to dehydrin. This link needs to be established.

6.       Is there enough compelling evidence to expect the collinearity between one protein structure and function or phenotypic trait? What if multiple proteins are involved in this stress-responsive behavior?

7.       I am a bit worried about scientific premise here, is DHN the only salinity biomarker? Also, wouldn’t drought and salinity stress go hand in hand? Does the drought stress also have the same biomarkers?

Materials & Mtds:

1.       Is it DNH or DHN genes? Please be consistent.

Results:

1.       Figures are not very clear. Please make it clearer and more legible. The fonts can be increased for clarity.

2.       Authors state the sequence differences in DHN segments, but has it been established that these are functionally essential segments? Does protein modelling show any topological changes that can be observed when comparing the protein through all 14 species. I think there should be a comment about this.

3.       Also, it will be important to discuss how these sequence variations can result in functional changes.

Conclusion:

How will this work help or pave the way for future research? How will this info be used for practical application of this research will be in future – there needs to be a comment about it.

Author Response

-- We would like to thank the reviewer for the tremendous efforts and important comments and suggestions to improve the manuscript.

Abstract:

  1. There are lot of spelling mistakes typos. Please proof-read.

-- The entire manuscript was revised and linguistic issues were resolved

  1. SK3-type what? It needs to be specified.

--  the SK3 type was clarified to the by reordering the sentence.

  1. The examples of S-segment variations are not needed in abstract in my opinion.

-- You are right, they are too many, the four examples were reduced to two

Introduction:

  1. Make sure all scientific names are italicized.

-- Thank you for your comment, all scientific names are italicized as requested.

  1. Some sentence structures are awkward and not clear due to the use of incorrect conjunctions. Please fix these.

-- The entire manuscript was revised and linguistic issues were resolved

  1. What happens in ERD? Is it a phase or response where genes are activated due to stress. Need to clarify this and tie it to the OSCA gene. Is OSCA the only gene that is affected by salinity? Or is it drought? or both?

-- Thank you for this comment. ERD is the early name of the protein, and the new name is OSCA. And yes it is not the only gene expressed in salinity but it is one of the first steps sensing the stress. And you are right it is not expressed in salinity stress only but also under drought. All these issues were expressed in revised form in more details.

  1. Is 150 mM NaCl equivalent to salinity in the environment?

-- Yes, it is equivalent to 13 ds/m, which is normal in soils from several regions in Jordan including Azaq.

  1. Are OSCA a type of dehydrin? There is no connection built in jumping from OSCA to dehydrin. This link needs to be established.

-- Again thanks for the comment. OSCA is the first protein to sense both drought and salinity stresses

  1. Is there enough compelling evidence to expect the collinearity between one protein structure and function or phenotypic trait? What if multiple proteins are involved in this stress-responsive behavior?

-- You are right, but this is the central protein sequence to unique function basis for all proteins.

  1. I am a bit worried about scientific premise here, is DHN the only salinity biomarker? Also, wouldn’t drought and salinity stress go hand in hand? Does the drought stress also have the same biomarkers?

-- You are right, there are several other biomarker for salinity and some would overlap with drought and some do not, please see our cited reference number [16]. This was added too.

Materials & Mtds:

  1. Is it DNH or DHN genes? Please be consistent.

-- You are right, they were all replace to DHN

Results:

  1. Figures are not very clear. Please make it clearer and more legible. The fonts can be increased for clarity.

-- Thank you for your important suggestion. Both figures 2 and 3 were replaced with high quality figures as requested.

  1. Authors state the sequence differences in DHN segments, but has it been established that these are functionally essential segments? Does protein modelling show any topological changes that can be observed when comparing the protein through all 14 species. I think there should be a comment about this.

-- Not at all, this needs to be investigated using directed mutagenesis to resolve this issue.

  1. Also, it will be important to discuss how these sequence variations can result in functional changes.

-- A new paragraph was introduced to the discussion section. The importance of DHN sequence as related to phosphorylation and DNA binding were discussed and related literature were cited.

Conclusion:

How will this work help or pave the way for future research? How will this info be used for practical application of this research will be in future – there needs to be a comment about it.

-- thank you for your important comment. The requested issue was highlighted in the conclusion section.

Round 2

Reviewer 1 Report

Authors have addressed my concerns in the revision.

Reviewer 2 Report

NA